# Investigating the Effectiveness of HyperTuning via Gisting

## Abstract

Gisting (Mu et al., 2023) is a simple method for training models to compress information into fewer token representations using a modified attention mask, and can serve as an economical approach to training Transformer-based hypernetworks. We introduce HyperLlama, a set of Gisting-based hypernetworks built on Llama-2 models that generates task-specific soft prefixes based on few-shot inputs. In experiments across P3, Super-NaturalInstructions and Symbol Tuning datasets, we show that HyperLlama models can effectively compress information from few-shot examples into soft prefixes. However, they still underperform multi-task fine-tuned language models with full attention over few-shot in-context examples. We also show that HyperLlama-generated soft prefixes can serve as better initializations for further prefix tuning. Overall, Gisting-based hypernetworks are economical and easy to implement, but have mixed empirical performance.

## 1 Introduction

With the increasing capability and popularity of large language models (LLMs), much recent work on LLMs has focused on improving the efficiency of training and serving models, such as via more efficient attention implementations (Dao et al. 2022, Dao 2023), lower precision operations (Dettmers et al., 2023; Frantar et al., 2023) and speculative decoding (Leviathan et al., 2023). Despite great advances in parameter-efficient fine-tuning (PEFT) methods and implementation improvements, fine-tuning a large language model to adapt it to a given downstream application still remains a computationally expensive task. In particular, parameter-efficient fine-tuning still generally requires backpropagating gradients through most of the model, despite only a small fraction of parameters requiring updates.

Given the strong in-context learning of LLMs (Brown et al., 2020), some recent work has explored using trained language models as optimizers–explicitly using their in-context learning capability to learn patterns or encode knowledge into hidden states or parameters (von Oswald et al., 2023a;b). Building on similar ideas, Phang et al. (2023) introduced *HyperTuning*: using specially trained large language as hypernetworks (Ha et al., 2017) to generate parameters from in-context inputs, without the need for backpropagation or gradient-descent-based training.

In this work, we investigate Gisting (Mu et al., 2023) as an approach for efficiently training models to compress few-shot examples into Gist token representations. Through large-scale multi-task training, Gisting models can serve as hypernetworks that generate task-specific soft-prefixes. Using LLaMA-2 (Touvron et al., 2023) models as the foundation, we train **HyperLlama**, a set of Gisting-based hypernetworks that generates soft prefixes based on few-shot examples. In experiments across P3 (Sanh et al., 2022), Super-NaturalInstructions (Wang et al., 2022) and Symbol Tuning datasets (Wei et al., 2023), we show that while HyperLlama models can effectively compress information from few-shot examples into a downstream model, their performance still generally pales in comparison to models with full attention over examples and trained for few-shot learning. We also show that the performance of Gisting-based hypernetworks can be boosted by performing an addition phase of pretraining to train the underlying model to perform Gisting, and jointly adapting the downstream model during the hypernetwork training phase. Consistent with prior work, we find that HyperLlama-generated soft prefixes can serve as better initializations for further prefix tuning, achieving better scores on held-out tasks.

Our contributions are as follows:

- We introduce HyperLlama, a Gisting-based hypernetwork that generates soft prefixes for a frozen downstream Llama-2 model.

- We show in experiments across P3, Super-NaturalInstructions and Symbol Tuning datasets that HyperLlama can effectively compress information from few-shot examples into soft prefixes, outperforming baselines that do not have access to additional examples. However, HyperLlama still generally underperforms Llama-2 models with full access to few-shot examples.

- We also show that HyperLlama-generated soft prefixes serve as better initializations for further prefix tuning.

## 2 RELATED WORK

**Hypernetworks and LLMs**   Hypernetworks were initially introduced by Ha et al. (2017) in the context of RNNs for sequence tasks. Recent work on hypernetworks have focused on using large foundation models (Bommasani et al., 2022) to generate a subset of parameters of a larger frozen model; this both significantly reduces the output space of the hypernetwork, and allows the resulting parameters to take advantage of an already capable pretrained model. Karimi Mahabadi et al. (2021) and He et al. (2022) trained hypernetworks as part of a larger model to perform efficiently cross-trak transfer learning. Phang et al. (2023) introduced HyperTuning, using a T5-based hypernetwork to generate soft prefixes and LoRA parameters from few-shot examples for a frozen downstream T5 model. Ivison et al. (2023) concurrently explored a T5-based hypernetwork with a similar setup, except also allowing the downstream model to access the hypernetwork encoder representations. Deb et al. (2022) similarly explored training BART models to generate parameters from task instructions. Outside of language models, Ruiz et al. (2023) trained a hypernetwork to generate a reduced version of LoRA parameters for personalizing generation of images of faces.

**Meta Learning and LLMs as Optimizers**   Large language models have been shown to be able to learn at inference time from in-context examples, through regular language modeling pretraining (Brown et al., 2020) or being explicitly trained to do in-context learning (Min et al., 2022; Shi et al., 2023). von Oswald et al. (2023a;b) showed that the attention mechanism can mimic gradient descent while processing in-context tokens, providing another approach to using Transformers to generate parameters or parameter updates.

**Parameter-Efficient Fine-tuning (PEFT) and Gisting**   To reduce memory and computation requirements for fine-tuning, many parameter-efficient fine-tuning methods have been proposed in recent years. We refer the reader to Lialin et al. (2023) for a comprehensive overview of PEFT methods. QLoRA (Dettmers et al., 2023), a method for fine-tuning LoRA (Hu et al., 2022) parameters against 4-bit quantized language models, is heavily used in this work to reduce the memory footprint of training runs. Gisting (Mu et al., 2023) involves training a decoder-only Transformer to compress information from earlier tokens into Gist token representations via appropriate modification of the attention mask. These Gist token representations are exactly equivalent to soft prefixes (Li & Liang, 2021), as discussed in Section 3.1.

## 3 HYPERTUNING WITH LANGUAGE MODELS

Parameter-efficient fine-tuning involves modifying a small number of parameters in a larger pretrained model through standard gradient descent-based training. The goal of *hypertuning* is to use a large language model to generate the corresponding modified parameters (e.g. soft prefixes, LoRA weights) in a single forward pass through a hypernetwork; those parameters can then be inserted in the frozen downstream model in the same configuration as in parameter efficient fine-tuning. The hypernetwork typically takes either few-shot examples and/or a task instruction as the input, and is generally initialized from the same parameters as the downstream model.

We briefly recap here the framework for hypertuning from Phang et al. (2023).

In standard fine-tuning, given a dataset of $N$ $(x, y)$ input-output pairs, a model $M$ with parameters $\theta$, and a loss function $\mathbb{L}$, we fine-tune $\theta$ based on the following objective:

$$\arg\min_{\theta} \frac{1}{N} \sum_{\{(x,y)\}} \mathbb{L}\Big(y, M(\theta; x)\Big) \tag{1}$$

During parameter-efficient fine-tuning (PEFT), instead of optimizing over $\theta$ directly, the majority of the parameters are frozen (represented by $\theta_0$), while only a much smaller subset of parameters $\phi$[1] are fine-tuned with the following objective.

$$\arg\min_{\phi} \frac{1}{N} \sum_{\{(x,y)\}} \mathbb{L}\Big(y, M(\theta_0; x, \phi)\Big) \tag{2}$$

HyperTuning introduces a hypernetwork $H$[2] that generates parameters $\hat{\phi}$ based on a set of K few-shot examples $\{(x_i, y_i)\}_K$, rather than directly optimizing over $\phi$. The hypernetwork is itself parameterized by $\xi$.

$$\hat{\phi} = H\Big(\xi; \{(x_i, y_i)\}_K\Big) \tag{3}$$

To train the hypernetwork $H$, we optimize over the hypernetwork parameters $\xi$. Combining Equations 2 and 3, we form the hypertuning training objective:

$$\arg\min_{\xi} \frac{1}{N} \sum_{\substack{\{(x,y)\} \\ \{\{(x_i,y_i)\}_K\}}} \mathbb{L}\Big(y, M(\theta_0; x, H(\xi; \{(x_i, y_i)\}_K))\Big) \tag{4}$$

As long as both $M$ and $H$ are differentiable, as is typically the case with neural networks, this optimization over $\xi$ can be performed simply with gradient descent.

## 3.1 HYPERTUNING WITH GISTING

In Phang et al. (2023), the authors introduced HyperT5, a set of T5-based hypernetworks that generated soft prefixes and LoRA (Hu et al., 2022) parameters based on few-shot examples. One of the major challenges identified in that work was the need to train parameter-generation heads, which were themselves large neural network modules which required significant training, necessitating an expensive hyperpretraining phase to initialize those modules before multi-task fine-tuning could be performed.

As additional background: Prefix tuning (Li & Liang, 2021) is a popular parameter-efficient fine-tuning method that involves directly fine-tuning a set of soft prefixes—key and value hidden states corresponding to a set of prefix tokens—to adapt a frozen language model to a downstream task. Phang et al. also found that HyperT5 performed better when trained to generate soft prefixes compared to generating LoRA parameters.

Recently, Mu et al. (2023) introduced Gisting, a simple method for fine-tuning a decoder Transformer to generate representations for Gist tokens that condense information from prior tokens. Referring to Figure 1a, by modifying the causal attention mask such that Gist tokens can attend to prefix tokens and suffix tokens can only attend to Gist tokens, a Gisting model can be trained to compress information relevant to language modeling on the suffix from the prefix within the Gist token representations. We emphasize that Gist token representation can be reduced to just their key and value hidden states, since these are the only hidden states that subsequent tokens interact with. Correspondingly, prefix tuning involves tuning soft prefixes, the hidden representations of key and value hidden states prepended to input text tokens. Hence, a Gisting model can be seen as a model that is trained to generate soft prefixes, with the gist representations being equivalent to soft prefixes in practice.[3]

---

[1]$\phi$ may be parameters such as soft prefixes or LoRA weights.

[2]Phang et al. uses the term *hypermodel*, but we use the term *hypernetwork* in this work for consistency with the field.

[3]There is some subtlety around handling position encodings, which we elaborate on in Appendix B.

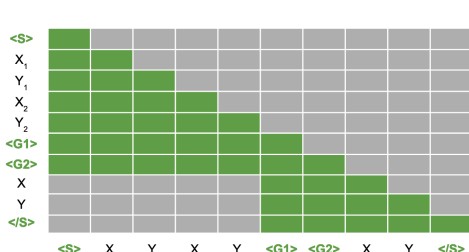 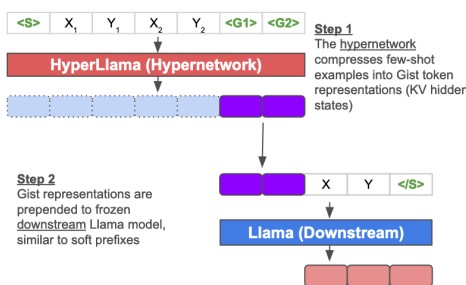

(a) **Gist Masking for HyperTuning**. The bottom left block of the causal attention mask is zeroed out, preventing tokens after the Gist tokens from attending to tokens before. Gist masking encourages the model to compress information from the few-shot examples into the Gist token hidden states, which the post-Gist tokens attend to.

(b) **Architecture of HyperLlama** HyperLlama consists of a *hypernetwork* that takes as input few-shot examples and generates task-specific soft prefixes (Gists) that get inserted into the *downstream model*. In practice, HyperLlama is a LoRA-tuned version of the Llama-2 model, so only one copy of the Llama-2 model weights needs to be kept in memory.

Compared to HyperT5, the advantage of using Gisting to generate soft prefixes is that it introduces no new layers or parameters (e.g. the parameter prediction heads in HyperT5), as Transformers already output key and value hidden states in regular operation. Gisting therefore provides a potentially highly economical approach to training LLM-based hypernetworks by building on already existing components in the Transformer architecture.

## 3.2 HYPERLLAMA: A GISTING-BASED HYPERNETWORK

We now introduce **HyperLlama**, a Gisting-based hypernetwork built on Llama-2 (Touvron et al., 2023). We illustrate the architecture of HyperLlama in Figure 1b. HyperLlama consists of two models, a *hypernetwork* that takes as input few-shot examples and compresses the information into Gist representations, and a *downstream model* that uses the Gist representations prepended to the input. The Gist representations thus effectively serve as soft prefixes. We use Llama-2-7B as the foundation for both the hypernetwork and downstream model, and use 16 Gist tokens for all experiments.

In practice, loading two large language models into memory is prohibitively expensive, especially during training. To reduce the computational cost of training the hypernetwork (which requires backpropagating gradients through both models), we apply QLoRA (Dettmers et al., 2023), quantizing both the hypernetwork and downstream model to 4-bit precision and using LoRA to modify the hypernetwork. During the forward pass through the hypernetwork, we activate the LoRA parameters for hypernetwork, and extract the Gist outputs. Then, we deactivate the LoRA parameters, restoring the untuned Llama-2 model, and feed both the Gist representations and the example inputs into the model. Aside from the LoRA parameters, we also introduce a set of additional embeddings for the Gist tokens. The LoRA parameters and the Gist embeddings are the only trained parameters of the hypernetwork.

We highlight that this approach differs from Mu et al. (2023), where they fine-tune all the model weights and use a single model that processes the pre-Gist tokens, the Gist tokens, and the post-Gist tokens.

## 3.3 COMPRESSION HYPERPRETRAINING

In some experiments, we follow Phang et al. (2023) and do an additional stage of pretraining to train the model to do perform Gisting before proceeding to multi-task fine-tuning. We refer to this as **compression hyperpretraining**, as it involves training the model to compress information into Gist tokens. The setup can be seen in Figure 2, and is based on the original hyperpretraining setup. Given a sequence of tokens from a pretraining corpus, we divide it into 4 segments: A, B, C and D. A and D are concatenated and serve as inputs to be compressed into the Gist representations <G>.

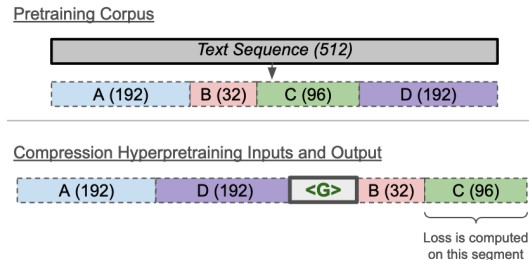

Figure 2: **Compression Hyperpretraining.** The model compresseses information in segments A and D into the Gist tokens `<G>`, and the model is trained to use that additional information to predict C based on B.

The additional context from A and D should help the model better predict tokens C from tokens B.[4] We use the RedPajama dataset (Computer, 2023) for hyperpretraining, and train for 50,000 steps. More training details are found in Appendix C.

Importantly, unlike with the training for hypertuning discussed above, we fine-tune all model parameters (i.e. no LoRA). This is the only step in this work that involves modifying the underlying Llama-2 parameters, and the compression-hyperpretrained models are only used in a subset of experiments, with the remainder using the base Llama-2 weights. We use FSDP (Zhao et al., 2023) to perform training in a memory-efficient manner.

### 3.4 FREEZING OR TUNING THE DOWNSTREAM MODEL

In most of the experiments in this work, the downstream model will either be a frozen base Llama-2, or a frozen Llama-2 after compression hyperpretraining. However, in specific experiments, we will also tune the downstream model. In those cases, we use a second set of LoRA weights for the downstream model.[5] We will make explicitly clear when the downstream model is also finetuned, referring to this as *downstream adaptation*. In all other cases, the downstream model is frozen during the multi-task training of the hypernetwork.

## 4 MULTI-TASK FINE-TUNING WITH HYPERLLAMA

We generally follow the training and evaluation recipe outlined in Phang et al. (2023). To evaluate the ability of LLM-based hypernetworks to generate task-specific parameters, we use the P3 (Sanh et al., 2022) and Super-NaturalInstructions (S-NI; Wang et al., 2022) datasets. Both datasets consist of a large number of in-domain training task datasets (62 for P3 and >1,000 for S-NI), and a smaller number of held-out task datasets for evaluation. Following the standard evaluation protocols for the respective datasets, evaluation for P3 held-out tasks is performed by accuracy from predicting the more likely answer option based on per-token logits, while for S-NI evaluation is performed via ROUGE on the generated responses. The hyperparameters for each experiment can be found in Appendix C.

### 4.1 MULTI-TASK FINE-TUNING ON P3

For P3, we use randomly sampled examples from the training set as the few-shot examples for input for the hypernetwork in the HyperLlama models and to be prepended to the input in the few-shot Llama-2B models. The 0-shot Llama-2 baseline is trained on single input-output pairs, similar to the T0 models. We find that the HyperLlama models generally outperform the 0-shot baseline but underperform the few-shot Llama-2 model, which is to be expected as the Llama-2 model can

---

[4]The entire BC sequence can actually be used as labels as in the standard language-modeling objective, but the separation of B and C is an artifact of following the hyperpretraining setup described in (Phang et al., 2023).

[5]In the actual implementation, we swap between the hypernetwork LoRA weights and the downstream LoRA weights depending on the phase of the forward pass.

|  | ANLI | HSwag | CB | COPA | RTE | WiC | WSC | WGD | AVG |
|---|---|---|---|---|---|---|---|---|---|
| *HyperTuning* | | | | | | | | | |
| HyperLlama-7B | 37.4 | 40.1 | 63.1 | 68.9 | 74.8 | 53.5 | 57.2 | 51.3 | 55.8 |
| HyperLlama-7B$^C$ | 38.2 | 42.2 | 61.8 | 76.1 | 74.0 | 54.6 | 56.4 | 51.1 | 56.8 |
| HyperLlama-7B$^D$ | 39.4 | 42.0 | 58.9 | 75.2 | 76.5 | 56.7 | 56.9 | 52.0 | 57.2 |
| HyperLlama-7B$^{CD}$ | 39.0 | 46.2 | 56.9 | 81.6 | 70.0 | 52.1 | 59.6 | 52.1 | 57.2 |
| *Fine-tuned LMs with In-Context Examples* | | | | | | | | | |
| Fine-tuned Llama-2-7B, 0-shot | 40.1 | 41.2 | 54.6 | 77.6 | 69.1 | 55.2 | 55.1 | 50.5 | 55.4 |
| Fine-tuned Llama-2-7B, 16-shot | 40.7 | 41.8 | 80.4 | 83.3 | 75.6 | 54.9 | 58.3 | 54.9 | 61.2 |
| T0-3B (Sanh et al., 2022) | 33.4 | 27.3 | 45.4 | 73.1 | 64.5 | 50.7 | 65.0 | 51.0 | 51.3 |

Table 1: **Results on P3 on held-out tasks with LLaMA-2 7B models.** For the HyperLlama models, $^C$ indicates hyperpretraining, while $^D$ indicates downstream adaptation.

directly attend to the few-shot examples and no compression is required. We also include results with variants of HyperLlama-2, including with compression hyperpretraining before the P3 training, and with downstream adaptation during P3 training. We find that both additions lead to improved performance, although used jointly there is no performance benefit.

## 4.2 MULTI-TASK FINE-TUNING ON SUPER-NATURALINSTRUCTIONS (S-NI)

We use only the English tasks in S-NI, which comprises over 1,000 task datasets. Each task includes a task definition as well as a set of preset positive examples for in-context learning, in addition to the actual examples of the task. For HyperLlama, we include both the definition and positive examples in the hypernetwork input, and only the actual task input for a given example as the downstream model input. We also train several other variants of HyperLlama on the S-NI dataset, incorporating compression hyperpretraining, downstream adaptation, and also additionally including the task definition in the downstream model input as well. We compare to baselines of training and evaluating Llama-2-7B with only the task definition, or task definition and few-shot examples as inputs (in addition to the actual task example input).

Results are shown in Table 2. We find that all versions of HyperLlama outperform the Llama-2 model provided only the task definition, demonstrating that HyperLlama is able to provide some useful information to the downstream model beyond just the task definition. All HyperLlama variants underperform the Llama-2 model with both the definition and few-shot examples, which is to be expected as discussed above. We also find that including the instruction in the downstream model (in addition to the hypernetwork) leads to a marked improvement in performance. This demonstrates that providing the downstream model with tokens it can attend to explicitly as opposed to through the hypernetwork can make a huge difference in performance, so the distribution of inputs between the hypernetwork and downstream model deserves significant consideration. Additionally, we find that incorporating both compression hyperpretraining and downstream adaptation leads to improved performance.

|  | AVG |
|---|---|
| *HyperTuning* | |
| HyperLlama-7B | 40.2 |
| HyperLlama-7B$^C$ | 41.7 |
| HyperLlama-7B$^{CI}$ | 46.4 |
| HyperLlama-7B$^{CDI}$ | 48.2 |
| *Fine-tuned LMs with In-Context Definitions+Examples* | |
| Fine-tuned Llama-2-7B (Def) | 39.7 |
| Fine-tuned Llama-2-7B (Def + Few-shot) | 52.7 |

Table 2: **Results on Super-NaturalInstuctions (S-NI) for Llama-7B models.** For the HyperLlama models, $^C$ indicates hyperpretraining, $^D$ indicates downstream adaptation, $^I$ indicates including instruction in the downstream model.

# 5 SYMBOL-TUNING WITH HYPERLLAMA

| Model | SUBJ | TEH | TEAB | TEAT | TEFE | TEHI | ADEC | OR | SOT | TOS | TC | AVG |
|---|---|---|---|---|---|---|---|---|---|---|---|---|
| *HyperTuning* | | | | | | | | | | | | |
| HyperLlama-7B | 51.0 | 49.6 | 35.3 | 37.5 | 36.3 | 39.0 | 54.4 | 47.6 | 50.4 | 63.8 | 55.4 | 47.3 |
| HyperLlama-7B$^C$ | 48.4 | 48.9 | 36.1 | 38.3 | 35.8 | 39.4 | 55.8 | 44.4 | 54.2 | 65.2 | 57.6 | 47.6 |
| HyperLlama-7B$^{CD}$ | 49.5 | 51.7 | 37.0 | 38.7 | 34.2 | 39.4 | 58.0 | 47.2 | 56.4 | 67.2 | 56.0 | 48.7 |
| *LMs with In-Context Examples* | | | | | | | | | | | | |
| Untuned Llama-2-7B, 0-shot | 49.3 | 50.8 | 34.5 | 32.7 | 34.8 | 33.0 | 51.4 | 51.4 | 36.2 | 49.2 | 49.4 | 43.0 |
| Untuned Llama-2-7B, 16-shot | 68.8 | 55.0 | 45.6 | 51.7 | 43.3 | 46.5 | 63.8 | 75.0 | 71.1 | 77.2 | 65.0 | 60.3 |
| Symbol-tuned Llama-2-7B, 16-shot | 79.5 | 57.6 | 47.3 | 51.2 | 49.7 | 48.4 | 68.6 | 88.4 | 58.4 | 70.6 | 83.6 | 63.9 |

Table 3: **Results on symbol tuning held-out tasks with HyperLlama and Llama-2 models.** HyperLlama models are trained with symbol tuning data. For the HyperLlama models, $^C$ indicates hyperpretraining, while $^D$ indicates downstream adaptation. HyperLlama models only slightly outperform the 0-shot Llama-2 baseline, while greatly underperforming the 16-shot Llama-2 baseline without any symbol tuning.

Symbol tuning (Wei et al., 2023) is a variant of instruction tuning (Sanh et al., 2022; Wei et al., 2022), where the natural language labels are swapped for random strings (random words, or strings of letters or numbers). For instance, all instances of 'Apple' may be substituted for 'True' and '7832' substituted for 'False'. Wei et al. showed that symbol tuning LLMs improves their in-context learning behavior and allows them to generalize better to tasks without instructions or natural language labels. The symbol tuning experimental setup is of interesting to us because it bypasses an issue with evaluating on the P3 and S-NI datasets above, where the task inputs cannot easily be separated from the instructions. In contrast, the symbol tuning training and evaluation sets include a variant where task instructions are omitted and the labels are swapped for irrelevant strings, allowing us to purely isolate the ability for either a hypernetwork or in-context learning to acquire knowledge of a task.

To evaluate the models on symbol tuned data, we use the same set of eleven held-out tasks with swapped labels and no instructions as in the original work. Because labels are randomly sampled even at evaluation time, we average over 10 random seeds for evaluating each model. All evaluation is computed with accuracy, with predictions based on the relative probabilities of output strings as with the P3 experiments.

We make two modifications compared to the original setup. First, whereas Wei et al. performed symbol tuning on instruction-tuned Flan-PaLM models, we perform symbol tuning on Llama-2 or HyperLlama models that have not been trained on instruction-formatted task data. Second, the original evaluation setup used a different distribution of random labels at evaluation time compared to training time to evaluate the generalization capability of symbol tuning. We found that this mismatch of distributions can greatly impact the performance of symbol-tuned models. We make a mild adjustment to the distribution of random labels in the evaluation distribution to make them more in line with the training distribution, in a manner that does not detract from the ability for models to generalize to unrelated label strings at inference time. More details on evaluation can be found in Appendix D.

For the HyperLlama models, we follow the setup in the above sections, performing few-shot examples of the symbol-tuning training data to the hypernetwork and one input-output pair to the downstream model. We likewise include results with the HyperLlama variants with hyperpretraining and downstream adaptation. We compare the performance of HyperLlama models to a 0-shot Llama-2 model, a few-shot prompted Llama-2 model, and a symbol-tuned Llama-2 model. We expect the 0-shot model to perform effectively at chance, as the output label string have no correspondence with actual output classes. The symbol-tuned Llama-2 model should serve as an upper-bound of hypernetwork performance, since the model has full access to the in-context examples.

Our results are shown in Table 3. We verify that 0-shot Llama-2 performs close to chance (the evaluation tasks are a mix of 2-class and 3-class classification tasks), while Llama-2 with symbol tuning performs the best. We find that the HyperLlama models perform quite poorly, slightly outperforming the 0-shot Llama-2 baseline, but severely underperforming even the few-shot Llama-2 model *without* symbol tuning. This suggests that, even in the absence of instructions and close associa-

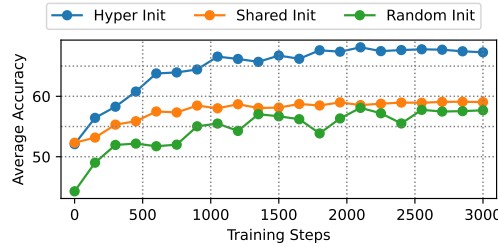

Figure 3: **Prefix tuning performance on P3 held-out tasks with different initializations**. HyperLlama-generated soft prefixes are better initializations for prefix tuning, achieving higher overall scores compared to baselines.

tions between output classes and label strings, the HyperLlama models still poorly take advantage of few-shot examples in the hypernetwork's inputs.

## 6 HYPERTUNING FOR IMPROVED PARAMETER INITIALIZATION

While HyperLlama is able to generate parameters from in-context examples, we expect the performance of the parameters to fall short of full parameter-efficient fine-tuning, given that it only sees a small set of in-context examples, and the parameters are generated with a single forward pass through the hypernetwork. However, the generated parameters can be used as initializations for further fine-tuning, and Phang et al. (2023) showed that HyperT5-generated parameters perform better as initializations compared to other baselines.

We conduct the same experiments as in Phang et al., performing prefix tuning with the HyperLlama-generated parameters on the P3 held-out tasks. We use HyperLlama-7B trained on P3 as the hypernetwork, and perform prefix tuning with the Llama-2 models.[6] We compare to two baselines: shared initialization (performing prefix tuning on the P3 training set, and using the resultant soft prefixes as the initialization), and random initialization of the soft prefixes. In these experiments, we perform prefix tuning without the reparameterization mentioned in Li & Liang (2021), and instead directly fine-tune the soft prefixes. This allows for the fairest comparison between the different initialization schemes, but also leads to the extremely poor performance of the random initialization, consistent with the findings of Li & Liang. We discuss this further in Appendix E.

The results are shown in Figure 3 and Table 4. Consistent with Phang et al., we find that the HyperLlama-generated soft prefixes outperform both the shared and random initializations, achieving higher scores throughout the fine-tuning process.

|  | ANLI | HSwag | CB | COPA | RTE | WiC | WSC | WGD | AVG |
|---|---|---|---|---|---|---|---|---|---|
| Hyper Init | 50.5 | 55.4 | 87.5 | 86.0 | 83.8 | 59.7 | 63.2 | 52.2 | 67.3 |
| Shared Init | 34.5 | 40.6 | 87.5 | 71.0 | 77.3 | 48.3 | 61.1 | 52.1 | 59.0 |
| Rand Init | 42.4 | 41.2 | 67.9 | 60.0 | 81.2 | 57.1 | 63.2 | 48.5 | 57.7 |

Table 4: **Full prefix tuning performance on P3 held-out tasks**.

## 7 DISCUSSION

We have seen above a set of mixed results from performing hypertuning via Gisting. Generally, the HyperLlama model is able to incorporate some information from the few-shot examples provided to the hypernetwork and impart useful information to the downstream model, thereby generally outperforming even a trained zero-shot model. However, we consistently find that HyperLlama still meaningfully underperforms Llama-2 models that have full access and attention over few-shot

---

[6]We perform prefix tuning without reparameterization for a fair comparison, as the HyperLlama only generated the flat prefix. Refer to Appendix E for more discussion.

examples, even when both HyperLlama and the Llama-2 models are trained on the same data. This shows that the process of compressing and transferring the information from the few-shot examples is generally still far from perfect.

From qualitative analysis of HyperLlama model mistakes, we found that the model particularly struggles on cases where the downstream model would benefit from being able to directly refer to or copy from the few-shot examples–for instance, tasks where there is a very specific output format that is only discoverable through few-shot examples. Such cases where there is a high sensitivity to exact strings in the few-shot examples appear to be where the HyperLlama models perform worse, whereas the Llama-2 models can directly attend to the few-shot examples. We find additional evidence for this in our symbol tuning experiments, where we find that HyperLlama underperforms even a Llama-2 without symbol tuning. In this case, even the few-shot learning capability inherent in the base Llama-2 trumps HyperLlama, which struggles with compressing unintuitive input-output pairings.

More generally, hypertuning also possesses several other weaknesses, such as only being able to take as input as many examples as can fit into a language model's limited context, needing to generate the parameters in a single, forward pass with fixed amounts of computation and potentially high sensitivity to the selection of few-shot input examples.

However, HyperLlama possesses other advantages too. Compressing relevant task-specific information into a soft prefix is significantly more economical at inference time than needing to recompute or store the hidden states of the whole set of few-shot examples. A soft prefix generally consists of such a small number of tokens worth of hidden states that the additional computation to attend to them is negligible relative to the rest of the input, and the storage is also more convenient given the small size. Moreover, we have shown that the HyperLlama-generated soft prefixes can be further fine-tuned to achieve better performance, providing another avenue of computation savings. Lastly, Gisting-based hypernetworks models can be efficiently trained and served since they only require a modification to the attention mask and LoRA weights (for swapping between hypernetwork and downstream model phases), which are common components in modern LLM infrastructure.

Overall, while we have found that Gisting-based hypernetworks such as HyperLlama face certain limitations and underperform fine-tuned, full-context language models, we see them as a promising and easy starting point for further investigations on LLM-based hypernetworks.

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

## A    APPENDIX

## B    HYPERLLAMA IMPLEMENTATION COMPLEXITIES

One complexity that arises with using Gist tokens is dealing with position encodings. Transformer-based language models such as Llama-2 tend to be highly sensitive to position encodings, and providing an input with an incorrect encoded position can lead to very poor results. Specifically, the issue is that for a given input to the downstream model, to prepend to Gist tokens as a soft prefix, we would need the Gist tokens to be encoded to a position before the input. However, the Gist tokens appear at the end of the Gisted input, meaning it is in a position very far from zero.

Mu et al. (2023) addressed this in their implementation of Gisting by always jointly processing the whole sequence of the Gist input, the Gist tokens and the post-Gist input. An alternative and more flexible solution is to modify the implementation of Llama-2 to accept a 'position offset', which pushes encoded positions of the post-Gist input down. While RoPE (?) is a relative position encoding, implementation-wise it be treated as a position embedding scheme when precomputing the rotary matrix, so this change is easy to implement. Hence, the output of the hypernetwork actually consists of two elements: first the Gists (KV hidden states corresponding to the Gist tokens), and secondly the offset corresponding to the position of the final Gist token, which is also how many position the downstream input token needs to be 'pushed down'.

This has also an additional benefit in allowing one to average over Gists of different sets of few-shot examples, which otherwise would not be possible if the Gist tokens were encoded to different positions. This can be implemented be providing an initial offset to the hypernetwork itself, with the offsets computed such that the resulting Gist tokens have the same position ending.

Lastly this implementation tweak is also helpful for Prefix Tuning, which similarly requires pushing the input tokens down by the number of prefix tokens.

## C    TRAINING DETAILS

For hyperpretraining, we train for 50,000 steps.For the HyperLlama, we use sequence length of 1024 for the hypernetwork and 384 for the downstream model. For Llama-2 models, we use a sequence length of 1024. For generation in S-NI, we use a maximum generation length of 128 (the generation length is in addition to the above sequence lengths). For prefix tuning, we use a sequence length of 384.

| | Hyperpretraining | P3 | S-NI | Symbol Tuning | Prefix Tuning |
|---|---|---|---|---|---|
| Learning Rate | 2e-5 | 2e-5 | 2e-5 | 2e-5 | 1e-43 |
| Batch Size | 128 | 128 | 128 | 32 | 32 |
| # Steps | 50K/30K | 10,000 | 3,000 | 4,000 | 3,000 |

Table 5: Training Hyperparameters

## D  SYMBOL TUNING DETAILS

**Label Distributions**  In the symbol tuning experimental setup, the symbol tuning training and evaluation phases use different distributions of random strings for swapping out labels. In particular, one component of the training distribution uses random 4-digit integers, while the evaluation distribution uses 5-digit integers. We should in initial experiments that performance was especially poor due to the mismatch in this distribution. Specifically: the Llama-2 tokenizer tokenizes each digit separately, and through the symbol-tuning training phase, the models learned that numerical outputs never exceeded 4-digit integers, and hence would assign near-0 probability to predicting 5-digit integers. Hence, we modify the evaluation distribution to similarly only include up to 4-digit integers. We do not modify the other components of the random string distribution other than the random numbers component. We expect that this modification might slightly advantage the hypernetwork models since the downstream model does not have direct access to the few-shot example labels, but our results still show that the hypernetworks significant underperform the comparable in-context learning baseline.

**Label Sampling**  For both training and evaluation, as sample a different set of labels for each actual example. The few-shot examples are constrained to have the same label mapping as each actual example.

## E  ADDITIONAL NOTES ON PREFIX TUNING

As discussed in Li & Liang (2021), directly fine-tuning the soft prefixes in a language model generally leads to very poor performance, or is highly sensitive to initializations and training hyperparameters. To address this, the authors introduce a reparameterization trick, where instead of fine-tuning the soft prefixes directly, the authors introduce an MLP that take a static input and outputs the soft prefixes. For reasons not yet well explored, this *significantly* improved the performance and stability of prefix tuning.

In the context of this work, HyperLlama generates soft prefixes from the attention mechanism of the model. We then directly fine-tune the generated soft prefixes, with generally good results. To fairly compare this to other forms of prefix tuning, we would need to find other baselines that also fine-tune a soft prefix directly. In the shared baseline, we perform prefix tuning over the P3 dataset with the reparameterization, then run the MLP and extract the resulting soft prefix–this soft prefix is then an appropriate point of comparison to the HyperLlama-generated soft prefixes.

On the flip side, we might consider using prefix tuning with reparameterization as a point of comparison. This would be an unfair comparison, as it involves significantly more parameters than just the soft prefixes. If we proceed with this knowledge, we have two experiments we can run. We show the results in Figure 4 and in Table 6 We find that standard prefix tuning with reparameterization outperforms directly fine-tuning the sof tprefix without reparameterization with any initialization scheme, including share initialization and HyperLlama-generated initializations. However, we emphasize that directly fine-tuning the soft prefix requires much fewer parameters, and is thus much less expensive than with the reparameterization.

We use the following P3 prompt formats for the prefix tuning experiments.

1. anli_GPT_3_style_r1

2. hellaswag_complete_first_then

3. super_glue_cb_GPT_3_style

|  | ANLI | HSwag | CB | COPA | RTE | WiC | WSC | WGD | AVG |
|---|---|---|---|---|---|---|---|---|---|
| Hyper Init | 50.5 | 55.4 | 87.5 | 86.0 | 83.8 | 59.7 | 63.2 | 52.2 | 67.3 |
| Shared Init | 34.5 | 40.6 | 87.5 | 71.0 | 77.3 | 48.3 | 61.1 | 52.1 | 59.0 |
| Rand Init | 42.4 | 41.2 | 67.9 | 60.0 | 81.2 | 57.1 | 63.2 | 48.5 | 57.7 |
| Rand (Reparam) Init | 67.5 | 77.9 | 94.6 | 87.0 | 88.1 | 71.0 | 82.1 | 50.4 | 77.3 |

Table 6: **Full prefix tuning performance on P3 held-out tasks**.

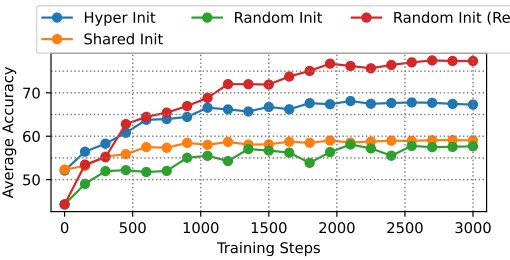

Figure 4: **Prefix tuning performance on P3 held-out tasks with different initializations, includ-ing reparameterization.**

4. super_glue_copa_C1_or_C2_premise_so_because_

5. super_glue_rte_GPT_3_style

6. super_glue_wic_GPT_3_prompt

7. super_glue_wsc.fixed_GPT_3_Style

8. winogrande_winogrande_debiased_Replace

