# OpenReview forum: "Investigating the Effectiveness of HyperTuning via Gisting"
_ICLR.cc/2025/Conference — Submitted to ICLR 2025_

### Official Review · Reviewer_76re · 2024-10-22

**Soundness:** 3
**Presentation:** 3
**Contribution:** 3
**Rating:** 6
**Confidence:** 4

**Summary:**

This work proposes a novel HyperLlama, which uses a hypernetwork to generate gist tokens to capture information from few-shot examples. The author provides sufficient experimental evidence that HyperLlama performs well on P3 and S-IN datasets, but performs not that good on Symbolic-tuning task. However, this work needs to strengthen its analysis of HyperLlama's performance deficiencies.

**Strengths:**

1. This paper proposes a novel method to generate gist tokens using hypernetwork, which can compress few shot examples of different tasks into gist tokens.
2. The author's motivation for designing this hypernetwork is very clear, and sufficient experiments are provided to prove the good performance of gist token on P3 and S-IN datasets, and its insufficient performance on Symbolic-tuning task.
3. The author has demonstrated through sufficient experiments that using the soft token generated by the hypernetwork as the initial token of prefix finetuning can improve the efficiency of prefix finetuning.

**Weaknesses:**

1. As noted in Section 3.3, "Compression Hyperpretraining" is essential for the downstream model (specifically the Llama 2 model in this study) to support gist tokens. However, it remains uncertain whether this additional pretraining step ("Compression Hyperpretraining"), following the standard pretraining, might adversely affect the model's performance on other tasks.
2. As discussed in Section 5, HyperLlama does not exhibit superior performance on the Symbolic-turing task. The paper lacks an in-depth analysis of the factors contributing to this shortcoming. Specifically, it is not clear whether the hypernetwork is unable to extract all relevant information from the few-shot examples, or if the gist tokens are insufficient in storing the information from the few-shot examples.
3. The study lacks an ablation analysis regarding the number of few-shot examples and the number of gist tokens. In other words, it is not explored whether increasing the number of few-shot examples, while keeping the number of gist tokens constant, would lead to an improvement in performance.

**Questions:**

Please refer to the Weakness section

---

### Official Review · Reviewer_5q1N · 2024-10-24

**Soundness:** 3
**Presentation:** 2
**Contribution:** 2
**Rating:** 5
**Confidence:** 3

**Summary:**

This paper introduced a Gisting-based hypernetworks. This hypernetwork is built on Llama-2 and is able to generate task-specific soft prefixes based on few-shot inputs. To do so, the authors use two LLMs. One acts as the hypernetwork and the other one acts as the downstream network. To enable training, the hypernetwork is trained with QLoRA (plus additional embeddings) to produce Gist tokens to append to the downstream network.

**Strengths:**

1, The figures are good and especially helpful for understanding the main idea of this paper.

2, This paper has detailed training details.

**Weaknesses:**

1, Table 1 has not been cited in the paper. What is its role in this paper?

2, At training, the hypernetwork and the downstream network are quantized to 4-bit. Are they full-precision when used in the test stage? If it is, how to solve the gap between the quantized model and the full-precision model?

3, To generate Gist tokens, two forwards with LLM are needed, which may incur high costs.

4, Even though I am not an expert in the area, I still recommend the author reorganize this paper for clear expression.

**Questions:**

Please answer the Weaknesses.

---

### Official Review · Reviewer_C8AB · 2024-11-02

**Soundness:** 2
**Presentation:** 2
**Contribution:** 2
**Rating:** 3
**Confidence:** 3

**Summary:**

The authors introduce HyperLlama, a Gisting-based hypernetwork designed to generate soft prefixes for a frozen downstream Llama-2 model. Their experiments across diverse datasets demonstrate that HyperLlama effectively compresses information from few-shot examples into soft prefixes, outperforming baselines that lack access to additional examples. Additionally, they show that HyperLlama-generated soft prefixes provide superior initializations for further prefix tuning.

**Strengths:**

1. The introduction of HyperLlama leverages Gisting to generate soft prefixes, compressing task-specific information efficiently.

2. Comprehensive experiments across multiple datasets demonstrate HyperLlama's strengths, especially in initializing soft prefixes for prefix tuning, showing improved performance over random initializations.

**Weaknesses:**

1. The paper lacks clarity in several areas, including the introduction of motivation, the description of the methodology, and the presentation of experimental results and figures.

2. The experiments are limited to the Llama-2-7B model, and there are no evaluations on newer models (such as Llama-3) or larger-scale models (like 13B), which limits the generalizability of the findings.

3. HyperLlama struggles with tasks that rely on precise output formats or highly contextual few-shot examples, affecting its generalizability.

4. The method’s effectiveness is contingent on Gisting’s ability to compress information accurately, which may vary across different types of tasks.

**Questions:**

Just like the weaknesses above.

---

### Official Review · Reviewer_TYJ2 · 2024-11-03

**Soundness:** 3
**Presentation:** 2
**Contribution:** 2
**Rating:** 5
**Confidence:** 3

**Summary:**

- This study introduces HyperLlama, a Gisting-based hypernetwork designed to generate soft prefixes for a frozen downstream Llama-2 model.
- Through experiments on P3, S-NI, and Symbol Tuning datasets, they demonstrate that HyperLlama can compress few-shot example information into soft prefixes,
- HyperLlama-generated soft prefixes also serve as strong initializations for further prefix tuning, supporting efficient fine-tuning.

**Strengths:**

The paper is written in a way that makes the field and methodology easy to understand.

**Weaknesses:**

The Discussion section describes the strengths and weaknesses of HyperLlama well, but these aspects are not fully addressed in the paper. For example, while it is mentioned that HyperLlama saves resources during inference, the authors should provide numerical evidence to support this.

I believe that the Gist (Mu et al., 2023) and the HyperTuning (phang et al. 2023) have both made substantial contributions to this field, and I have some concerns that this paper primarily builds on these approaches by combining them in its experiments.

**Questions:**

(1) What are the advantages of this paper's approach compared to existing Gist methods? Additionally, I would like to understand the benefits of separating the model components.
(2) As highlighted in the title, it would be helpful if the paper demonstrated the effectiveness of the method with numerical evidence compared to existing methods.

---

### Official Review · Reviewer_5q58 · 2024-11-08

**Soundness:** 3
**Presentation:** 3
**Contribution:** 1
**Rating:** 3
**Confidence:** 3

**Summary:**

This paper introduce a set of Gisting based hyper network called HyperLlama for generating soft prefix tokens for downstream tasks. The prefix tokens acts similar to few shot example in in-context learning. The experiments show that their HyperLlama is effective in generating soft prefix tokens, but they underperformed compared to multi-task fine-tuned models with attention to in-context examples.

**Strengths:**

1. The paper is easy to read.
2. The paper has a very detailed discussion on each experiment.

**Weaknesses:**

It seems that the paper has no technical contribution to the community. The introduced HyperLlama follows the setting of HyperTuning.
The paper introduced a set of new models (HyperLlama) for sure, but I do not think it is a valid technical contribution. Further, the introduced models generally underperform few-shot language models by large margins.

**Questions:**

In second paragraph of 3.2, the authors says they use QLoRA for hyper network training, but in the last paragraph of 3.3, the authors states they fine-tuned all model parameters and no LoRA is used. I am a bit confused.

---

### Meta-Review · Area_Chair_YUkh · 2024-12-19

**Metareview:**

This paper received ratings of 5, 6, 5, 3, 3, where the reviewers assigned mixed-to-low ratings, primarily citing concerns over limited novelty, weak empirical results, and lack of clarity regarding the practical benefits of the proposed approach.

The submission introduces HyperLlama, a Gisting-based hypernetwork that leverages modified attention mechanisms to generate task-specific soft prefixes for large language models (LLMs).

Strengths:
- The idea of using Gisting-based hypernetworks for economical computation is conceptually interesting.
- The proposed methodology has the potential to reduce computational costs during inference.

Area for improvement:
- Experimental results are not convincing: HyperLlama fails to achieve competitive performance in critical benchmarks compared to fine-tuned and in-context learning models.
- Unclear practical relevance: the benefits of using HyperLlama in realistic scenarios remain not clera due to its limited effectiveness.
- Presentation Issues: The manuscript lacks clarity in explaining certain methodological choices, especially around compression hyperpretraining and task-specific adaptation strategies.

While the proposed approach demonstrates some conceptual promise, its empirical weaknesses and unclear practical relevance outweigh its strengths. The authors are encouraged to further refine their methodology, improve its performance, and provide more compelling evidence of its utility in future work.

**Additional Comments On Reviewer Discussion:**

The discussion between authors and reviewers was active, with the authors addressing key points raised in the initial reviews. Reviewers acknowledged the effort made to clarify and contextualize the work.

The authors made a commendable effort to address reviewers' concerns, providing clarifications on the use of hypernetworks, evaluation metrics, and the limitations of their approach. Their willingness to engage with feedback reflects a positive and collaborative attitude.
While some responses offered valuable context, important explanations seem not sufficiently detailed. Additional empirical evidence or comparisons would have helped substantiate their claims.

The rebuttal and subsequent discussion did not result in a significant shift in the reviewers’ perspectives. Concerns about the novelty, practical contributions, and consistent underperformance of the approach persisted and remain unresolved. For instance, the reviewers highlighted the limited novelty of the proposed approach and its underperformance compared to baseline models with direct access to few-shot examples. The authors are encouraged to address these issues/concerns and consider a resubmission.

---

### Decision · Program_Chairs · 2025-01-22

Reject